



# Earth System Model Aerosol-Cloud Diagnostics Package (ESMAC Diags) Version 1: Assessing E3SM Aerosol Predictions Using Aircraft, Ship, and Surface Measurements

Shuaiqi Tang[1], Jerome D. Fast[1], Kai Zhang[1], Joseph C. Hardin[2], Adam C. Varble[1], John E. Shilling[1], Fan Mei[1], Maria A. Zawadowicz[3] and Po-Lun Ma[1]

[1]Pacific Northwest National Laboratory, Richland, WA, USA
[2]ClimateAI Inc, San Francisco, CA, USA
[3]Brookhaven National Laboratory, Upton, NY, USA

*Correspondence to*: Shuaiqi Tang (shuaiqi.tang@pnnl.gov)

**Abstract.** An Earth System Model (ESM) aerosol-cloud diagnostics package is developed to facilitate the routine evaluation of aerosols, clouds and aerosol-cloud interactions simulated by the Department of Energy's (DOE) Energy Exascale Earth System Model (E3SM). The first version focuses on comparing simulated aerosol properties with aircraft, ship, and surface measurements, most of them are measured in-situ. The diagnostics currently covers six field campaigns in four geographical regions: Eastern North Atlantic (ENA), Central U.S. (CUS), Northeastern Pacific (NEP) and Southern Ocean (SO). These regions produce frequent liquid or mixed-phase clouds with extensive measurements available from the Atmospheric Radiation Measurement (ARM) program and other agencies. Various types of diagnostics and metrics are performed for aerosol number, size distribution, chemical composition, CCN concentration and various meteorological quantities to assess how well E3SM represents observed aerosol properties across spatial scales. Overall, E3SM qualitatively reproduces the observed aerosol number concentration, size distribution and chemical composition reasonably well, but underestimates Aitken mode and overestimates accumulation mode aerosols over the CUS region, and underestimates aerosol number concentration over the SO region. The current version of E3SM struggles to reproduce new particle formation events frequently observed over both the CUS and ENA regions, indicating missing processes in current parameterizations. The diagnostics package is coded and organized in a way that can be easily extended to other field campaign datasets and adapted to higher-resolution model simulations. Future releases will include comprehensive cloud and aerosol-cloud interaction diagnostics.



## 1. Introduction

Aerosol number, mass, size, composition, and mixing state affect how aerosol populations scatter and absorb solar radiation and influence cloud albedo, amount, lifetime, and precipitation (Twomey, 1977; Albrecht, 1989) by acting as cloud condensation nuclei (e.g., Petters and Kreidenweis, 2007). However, there are still knowledge and measurement gaps on the physical and chemical mechanisms regulating the sources, sinks, gas-to-particle partitioning (e.g., secondary formation processes), and spatiotemporal distribution of aerosol populations. Consequently, the representation of the aerosol lifecycle and the interaction of aerosol populations with clouds and radiation in Earth system models (ESMs) still suffer from large uncertainties (Seinfeld et al., 2016; Carslaw et al., 2018), which impacts the ability of ESMs to predict the evolution of the climate system (IPCC, 2013).

To facilitate model evaluation and document the performance of parameterizations in ESMs, many modeling centers have developed standardized diagnostics packages. Some examples focus on meteorological metrics include the U.S. National Center of Atmospheric Research (NCAR) Atmospheric Model Working Group (AMWG) diagnostics package (AMWG, 2021), the U.S. Department of Energy (DOE) Energy Exascale Earth System Model (E3SM, Golaz et al., 2019) diagnostics (E3SM, 2021), the European Union (EU) Earth System Model Evaluation Tool (ESMValTool, Eyring et al., 2016), and the Program for Climate Model Diagnosis and Intercomparison (PCMDI) Metric Package (PMP, Gleckler et al., 2016). Some recent efforts focus on process-oriented diagnostics (POD) that are designed to provide insights into parameterization developments to address long-standing model biases. Maloney et al. (2019) summarizes the activities by the U.S. National Oceanic and Atmospheric Administration (NOAA) Modeling, Analysis, Prediction, and Projections program (MAPP) Model Diagnostics Task Force (MDTF) to apply community-developed PODs to climate and weather prediction models. Zhang et al. (2020) developed a diagnostics package that utilizes statistics derived from long-term ground-based measurements from the DOE Atmospheric Radiation Measurement (ARM) User Facility for climate model evaluation. Aerosol properties, however, are not included in these diagnostics packages.

The international collaborative AeroCom project (Myhre et al., 2013; Schulz et al., 2006) focuses on evaluation of aerosol predictions using available measurements and includes intercomparisons among global models to assess uncertainties in seasonal and regional variations in aerosol properties and their potential impact on climate. Their diagnostics heavily rely on satellite remote sensing products (e.g., aerosol optical depth) which have global coverage but poor spatial and temporal resolution that hinders a process-level understanding of the sources of model uncertainty. More recently, the Global Aerosol Synthesis and Science Project (GASSP, Reddington et al., 2017; Watson-Parris et al., 2019) has developed a global database of aerosol observations from fixed surface sites as well as ship and aircraft



platforms from 86 field campaigns between 1990 and 2015 that can be used for model evaluation. Recent
field campaigns after year 2015 are not included in this effort.
Many aerosol properties are difficult to measure directly. Remote sensing instruments (e.g., ground
and satellite radiometers) that only measure radiative properties of column-integrated aerosols, such as
optical depth, are frequently used to evaluate model predictions. Instruments such as ground lidars (e.g.,
Campbell et al., 2002) or lidars onboard aircraft (e.g., Müller et al., 2014) and satellite (e.g., CALIPSO,
Winker et al., 2009) platforms can provide vertical profiles of aerosol extinction, backscatter, and/or
depolarization, but they do not directly measure aerosol number, size and composition. Therefore, the
quantities measured by remote sensing instruments cannot be used alone to assess model predictions of
aerosol-radiation-cloud-precipitation interactions. Surface monitoring sites provide long-term in situ
aerosol property measurements but are limited to land locations with far fewer operational sites compared
to those dedicated to routine meteorological sampling. Ship and aircraft platforms are commonly
deployed during field campaigns to obtain in situ and remote sensing aerosol property measurements in
remote or poorly sampled locations such as over the ocean and within the free troposphere, which are
highly valuable when studying spatial variations of aerosols. Aircraft platforms also provide a means to
obtain coincident measurements of aerosol and cloud properties needed to understand their interactions.
Although in-situ ship and airborne aerosol measurements are usually limited to specific locations for short
time periods, the increasing number of completed field campaigns conducted over a range of atmospheric
conditions provides an opportunity to use them for model evaluation.
As noted by Reddington et al. (2017), the considerable investment in collecting field campaign
measurements of aerosol properties is underexploited by the climate modeling community. This can be
largely attributed to datasets located in disparate repositories and the lack of a standardized file format
that requires excessive time and effort spent on manipulating the datasets to facilitate comparisons
between observed and simulated values, especially for those unfamiliar with measurement techniques,
assumptions, and uncertainties. With many field campaigns conducted since 2015 being available but
rarely used for model evaluation, this study describes the first version of the ESM Aerosol-Cloud
Diagnostics (ESMAC Diags) package to facilitate the evaluation of ESM-predicted aerosols, utilizing
recent measurements from aircraft, ship and surface platforms collected by the U.S. DOE ARM and
National Science Foundation (NSF) NCAR user facilities, most of which are in-situ measurements. The
overall structure of ESMAC Diags is designed similar to the Aerosol Modeling Testbed for the Weather
Research and Forecasting (WRF) model described in Fast et al. (2011), except that it uses Python to
interface the measurements with ESM output and does not preprocess the observational dataset into a
common format. The diagnostics package is firstly designed with and applied to E3SM Atmosphere

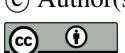



Model version 1 (EAMv1, Rasch et al., 2019). EAMv1 uses an improved modal aerosol treatment
implemented based on the 4-mode version of the modal aerosol module (MAM4, Liu et al., 2016), such
as improved treatment of $H_2SO_4$ vapor for new particle formation, improved SOA treatment, new MOA
species, improvements to aerosol convective transport, wet removal, resuspension from evaporation and
aerosol-affected cloud microphysical processes (Wang et al., 2020). Only minimal modifications to the
diagnostics package are needed for potential application to other ESMs.

**2.   Introduction of ESMAC Diags**
The diagnostics package is designed to be flexible so that additional measurements and functionality can
be included in the future. The workflow of ESMAC Diags, illustrated in Figure 1, consists of six major
components. The "scripts" directory contains executable scripts and user-specified settings. The "src"
directory contains all source code including code used to preprocess model output, read files, merge
measurements from different instruments, compute observed versus simulated statistical relationships,
and plot results. All observational and model data in the "data" directory are organized by field campaign.
The diagnostic plots and statistics are put in the "figures" directory, also organized by field campaign.
The "testcase" directory includes a small amount of input and verify data to test if the package is installed
properly. The "webpage" directory provides an interface to view diagnostics figures. It is relatively
straightforward to add other field campaigns or datasets using this structure. Most of the datasets used in
ESMAC Diags are in a standardized netCDF format (NETCDF, 2021); however, some ARM aircraft
measurements use different ASCII formats. Currently, the diagnostic package reads observational data
directly from their original format. In the long term, we may standardize the observational data format in
a similar manner as was done in GASSP project (Reddington et al., 2017).



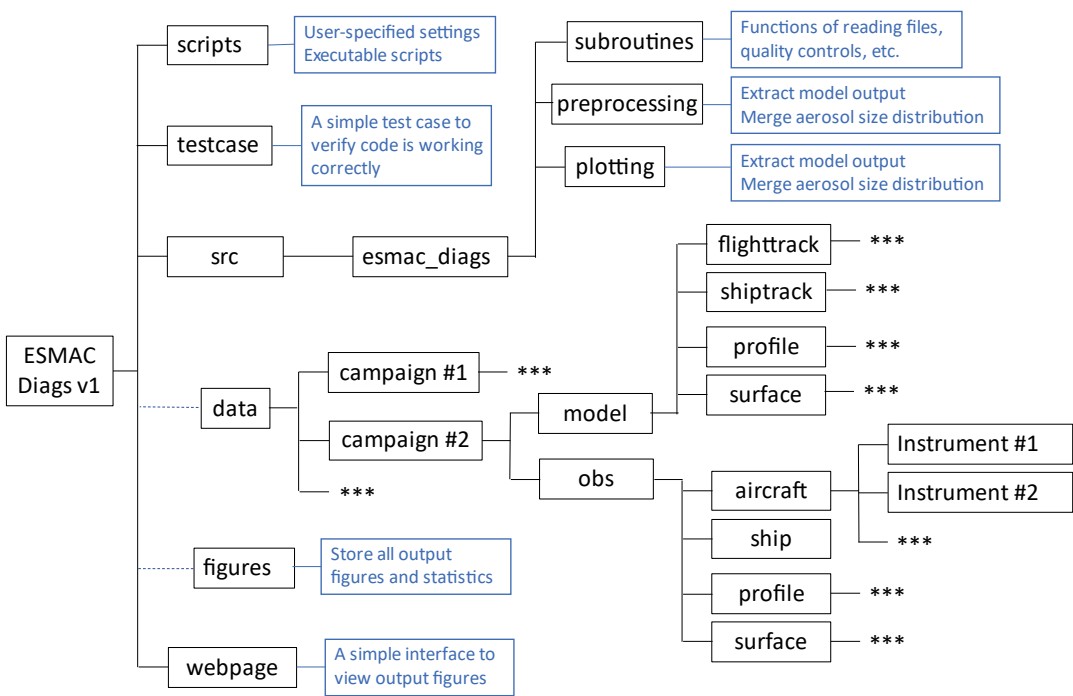


**Figure 1: Workflow of ESMAC Diags. Boxes in blue describe the functions of the directory. Asterisks represent boxes that follow the same format as those shown in parallel.**

### 2.1 Field observations and merged aerosol size distribution

We initially focus on four geographical regions where liquid clouds occur frequently and extensive measurements are available from ARM and other agencies: Eastern North Atlantic (ENA), Northeastern Pacific (NEP), Central U.S. (CUS, where the ARM Southern Great Plains, SGP, site is located), and Southern Ocean (SO). Aerosol properties also vary among these regions. Six field campaigns from these four testbeds are selected in the version 1.0 of ESMAC Diags (Table 1). HI-SCALE and ACE-ENA are based on long-term ARM ground sites with aircraft field campaigns sampling below, within, and above convective and marine boundary layer clouds, respectively, within a few hundred kilometers around the sites. CSET and MAGIC are field campaigns with aircraft and ship platforms, respectively, sampling transects between California and Hawaii characterized by a transition between stratocumulus and trade cumulus dominated regions. SOCRATES and MARCUS are field campaigns with aircraft and ship platforms, respectively, based out of Hobart, Australia. Aircraft transects during SOCRATES extended





south to around 60°S, while ship transects during MARCUS extended southwest from Hobart to
Antarctica. The aircraft (black) and ship (red) tracks for these field campaigns are shown in Figure 2.
**Table 1. Descriptions of the field campaigns used in this study. Numbers after aircraft or ship**
**represent number of flights or ship trips in each field campaign or IOP.**

| Campaign* | Period | Platform | Typical Conditions | Reference |
|---|---|---|---|---|
| **HI-SCALE** | IOP1: 24 Apr – 21 May 2016 IOP2: 28 Aug – 24 Sep 2016 | Ground, aircraft (IOP1: 17, IOP2: 21) | Continental cumulus with high aerosol loading | (Fast et al., 2019) |
| **ACE-ENA** | IOP1: 21 Jun – 20 Jul 2017 IOP2: 15 Jan – 18 Feb 2018 | Ground, aircraft (IOP1: 20, IOP2: 19) | Marine stratocumulus with low aerosol loading | (Wang et al., 2021) |
| **MAGIC** | Oct 2012 – Sep 2013 | Ship (18) | Marine stratocumulus to cumulus transition with low aerosol loading | (Lewis and Teixeira, 2015; Zhou et al., 2015) |
| **CSET** | 1 Jul – 15 Aug 2015 | Aircraft (16) | Same as above | (Albrecht et al., 2019) |
| **MARCUS** | Oct 2017 – Apr 2018 | Ship (4) | Marine liquid and mixed phase clouds with low aerosol loading | (McFarquhar et al., 2021) |
| **SOCRATES** | 15 Jan – 24 Feb, 2018 | Aircraft (14) | Same as above | (McFarquhar et al., 2021) |

* full names of the listed field campaigns:
HI-SCALE: Holistic Interactions of Shallow Clouds, Aerosols and Land Ecosystems
ACE-ENA: Aerosol and Cloud Experiments in the Eastern North Atlantic
MAGIC: Marine ARM GCSS Pacific Cross-section Intercomparison (GPCI) Investigation of Clouds
CSET: Cloud System Evolution in the Trades
MARCUS: Measurements of Aerosols, Radiation and Clouds over the Southern Ocean
SOCRATES: Southern Ocean Cloud Radiation and Aerosol Transport Experimental Study





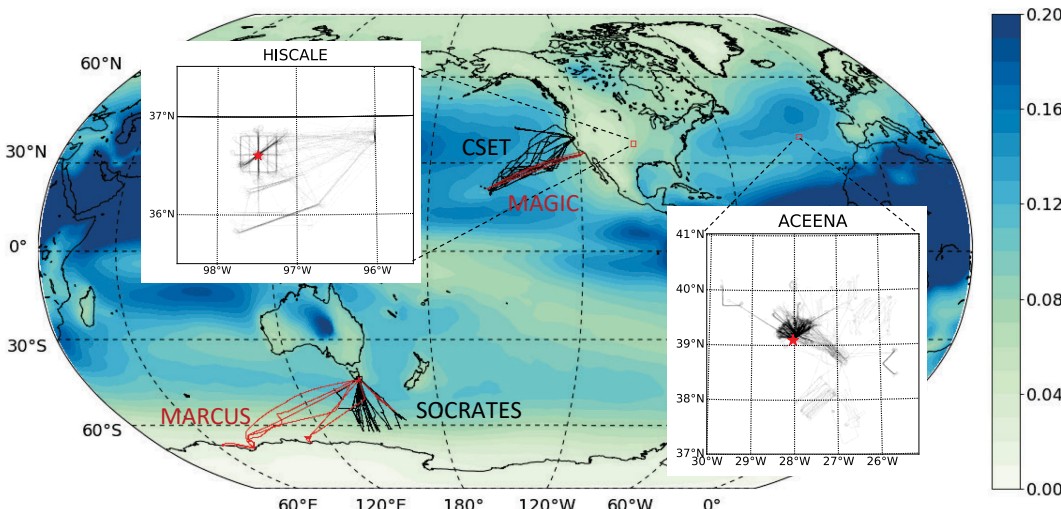

**Figure 2. Aircraft (black) and ship (red) tracks for the six field campaigns. Overlaid is aerosol optical depth at 550nm averaged from 2014 to 2018 simulated in EAMv1.**

The instruments and measurements used in ESMAC Diags version 1.0 are listed in Table 2. Note that some instruments are only available for certain field campaigns or failed operationally during certain periods, so that model evaluation is limited by the availability of data collected in each field campaign. ARM data usually include quality flags indicating bad or indeterminate data. These flagged data are filtered out, except surface CPC measurements for HI-SCALE, that data flagged as greater than maximum value (8000 cm$^{-3}$) are retained since aerosol loading can be higher than that during new particle formation events. This exception ensures a reasonable diurnal cycle shown in Section 3.3. For some data that do not have a quality flag (e.g., UHSAS data in NCAR research flight measurements), a simple minimum and maximum threshold is applied. For some field campaigns (HI-SCALE and ACE-ENA), there are several instruments (e.g., FIMS, PCASP, OPC for aircraft; SMPS and nanoSMPS for ground) measuring aerosol size distribution over different size ranges. These datasets are merged to create a more complete size distribution. The aerosol concentrations in the "overlapping" bins measured by multiple instruments are weighted by the uncertainty of each instrument based on the knowledge of the ARM instrument mentors. An example of the merged aerosol size distribution and individual measurements for one flight in ACE-ENA is shown in Figure 3. Ranging from $10^1$ to $10^4$ nm, the merged aerosol size distribution data account for ultrafine, Aitken, and accumulation modes.

**Table 2. List of instruments and measurements used in ESMAC Diags v1.0.**

| Instrument | Platform | Measurements | Available campaigns | DOIs or References |
|---|---|---|---|---|
| | | | | |





| Surface meteorological station (MET) | Ground, ship | Temperature, relative humidity, wind, pressure | HI-SCALE, ACE-ENA, MAGIC, MARCUS | HI-SCALE, ACE-ENA: (Kyrouac and Shi, 2018) MAGIC: (ARM, 2014) MARCUS: 10.5439/1593144 |
|---|---|---|---|---|
| Scanning mobility particle sizer (SMPS) | Ground | Aerosol size distribution (20-700 nm) | HI-SCALE | (Howie and Kuang, 2016) |
| Nano scanning mobility particle sizer (nanoSMPS) | Ground | Aerosol size distribution (2-150 nm) | HI-SCALE | (Koontz and Kuang, 2016) |
| Ultra-High Sensitivity Aerosol Spectrometer (UHSAS) | Ground, aircraft, ship | Aerosol size distribution (60 – 1000 nm), number concentration | HI-SCALE, ACE-ENA, MAGIC, MARCUS, CSET, SOCRATES | HI-SCALE, MAGIC, MARCUS: (Koontz and Uin, 2018) ACE-ENA: (Uin et al., 2018) CSET: 10.5065/D65Q4T96 SOCRATES: 10.5065/D6M32TM9 |
| Condensation particle counter (CPC) | Ground, aircraft, ship | Aerosol number concentration (> 10 nm) | HI-SCALE, ACE-ENA, MAGIC, MARCUS | HI-SCALE (ground): (Kuang et al., 2016) ACE-ENA (ground), MAGIC: (Kuang et al., 2018a) MARCUS: (Kuang et al., 2018b) HI-SCALE (aircraft): (ARM, 2016b) ACE-ENA (aircraft): (Mei, 2018) |
| Condensation particle counter – ultrafine (CPCU) | Ground, aircraft | Aerosol number concentration (> 3 nm) | HI-SCALE, ACE-ENA | HI-SCALE (ground): 10.5439/1046186 HI-SCALE (aircraft): (ARM, 2016b) ACE-ENA (aircraft): 10.5439/1440985 |
| Condensation nuclei counter (CNC) | Aircraft | Aerosol number concentration (11-3000 nm) | CSET, SOCRATES | CSET: 10.5065/D65Q4T96 SOCRATES: 10.5065/D6M32TM9 |
| Cloud condensation nuclei (CCN) counter | Ground, aircraft, ship | CCN number concentration (0.1% to 0.5% supersaturation* depending on the platform) | HI-SCALE, ACE-ENA, MAGIC, MARCUS, SOCRATES | HI-SCALE (ground), ACE-ENA (ground), MARCUS: 10.5439/1342133 MAGIC: 10.5439/1227964 SOCRATES: 10.5065/D6Z036XB HI-SCALE (aircraft): (ARM, 2016a) |
| Aerosol chemical speciation monitor (ACSM) | Ground | Aerosol composition | HI-SCALE, ACE-ENA | 10.5439/1762267 |
| Microwave radiometer (MWR) | Ground, ship | Liquid water path, precipitable water vapor | MAGIC, MARCUS | 10.5439/1027369 |
| Counterflow virtual impactor (CVI) | Aircraft | Separates large droplets or ice crystals | HI-SCALE, ACE-ENA, SOCRATES | HI-SCALE: (ARM, 2016a) ACE-ENA: 10.5439/1406248 SOCRATES: 10.5065/D6M32TM9 |
| Fast integrated mobility | Aircraft | Aerosol size distribution (10 – 425 nm) | HI-SCALE, ACE-ENA | HI-SCALE: (ARM, 2017) ACE-ENA: (ARM, 2020) |



| spectrometer (FIMS) | | | | |
|---|---|---|---|---|
| Passive cavity aerosol spectrometer (PCA SP) | Aircraft | Aerosol size distribution (120 – 3000 nm) | HI-SCALE, ACE-ENA, CSET | HI-SCALE: (ARM, 2016a) ACE-ENA: (ARM, 2018) CSET: 10.5065/D65Q4T96 |
| Optical particle counter (OPC) | Aircraft | Aerosol size distribution (390 – 15960 nm) | ACE-ENA | (ARM, 2018) |
| Interagency working group for airborne data and telemetry systems (IWG) | Aircraft | Navigation information and atmospheric state parameters | HI-SCALE, ACE-ENA | HI-SCALE: (ARM, 2017) ACE-ENA: (ARM, 2018) |
| High-resolution time-of-flight aerosol mass spectrometer (AMS) | Aircraft | Aerosol composition | HI-SCALE, ACE-ENA | HI-SCALE: (ARM, 2017) ACE-ENA: 10.5439/1468474 |
| Water content measuring system (WCM) | Aircraft | Cloud liquid and total water content | HI-SCALE, ACE-ENA | HI-SCALE: (ARM, 2016a) ACE-ENA: 10.5439/1465759 |
| Doppler lidar (DL) | Ground | Boundary layer height | HI-SCALE | 10.5439/1726254 |

* for measured supersaturations (SS) that vary over time, a ± 0.05% window is applied (e.g., 0.5% SS
includes samples with SS between 0.45% and 0.55%).

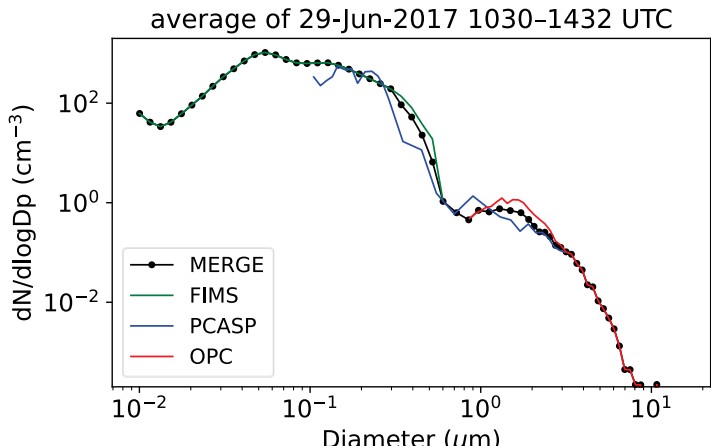


**Figure 3. An example of a mean aerosol number distribution merged from FIMS, PCASP and OPC**
**instruments for ACE-ENA aircraft measurements on 29 June 2017.**



## 2.2 Preprocessing of model output

We configured the EAMv1 to follow the Atmospheric Model Intercomparison Project (AMIP) protocol (Gates et al., 1999) with real-world forcings (e.g., greenhouse gases, sea surface temperature, aerosol emissions, etc.). In this study, we run the model from 2012 to 2018, covering all six field campaign periods introduced previously, with at an additional 10 months for model spin-up. For each simulation year, we use the year 2014 emission data from CMIP6, since the emission data does not cover years after 2014. The simulated horizontal winds are nudged towards the Modern-Era Retrospective analysis for Research and Applications, Version 2 (MERRA-2, Gelaro et al., 2017) with a relaxation time scale of 6 hours. Previous studies (Sun et al., 2019; Zhang et al., 2014) showed that with such nudging configuration the large-scale circulation is well constrained in the nudged simulation, especially for the mid- and high-latitude regions. The simulation uses a horizontal grid spacing of ~1° (NE30, the number of elements along a cube face of the E3SM High-Order Methods Modeling Environment, HOMME, dynamics core) with a 30-minute timestep. We saved hourly output to compare with field campaign measurements. The diagnostics package post-processes 3-D model variables associated with aerosol concentration, size, composition, optical properties, precursor concentration, CCN concentration, and atmospheric state variables. The size of output data is reduced by saving 3-D variables only over the field campaign regions.

We extracted model output along the aircraft (ship) tracks using an "aircraft simulator" (Fast et al., 2011) strategy to facilitate comparisons of observations and model predictions. At each aircraft (ship) measurement time, we find the nearest model grid cell, output time slice, and vertical level of the aircraft altitude (or the lowest level for ship) to obtain the appropriate model values. Since there are both spatial and temporal mismatch existing between model output and field measurements, the evaluation focuses on overall statistics. We also calculate the aerosol size distribution from 1 nm to 3000 nm at 1 nm increments from the individual size distribution modes in MAM4 to facilitate comparisons with observed aerosol number distribution that has different size ranges for different instruments. All these variables are saved in separate directories according to the specific aircraft (ship) tracks, as indicated in Figure 1.

## 2.3 List of diagnostics and metrics

Currently, ESMAC Diags produces the following diagnostics and metrics:

- Mean value, bias, RMSE and correlation of aerosol number concentration.
- Timeseries of aerosol variables (aerosol number concentration, aerosol number size distribution, chemical composition, CCN number concentration) for each field campaign or intensive observational period (IOP) at the surface or along each flight (ship) track.

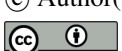



• Diurnal cycle of aerosol variables at the surface.
• Mean aerosol number size distribution for each field campaign or IOP.
• Percentiles of aerosol variables by height for each field campaign or IOP.
• Percentiles of aerosol variables by latitude for each field campaign or IOP.
• Pie/bar charts of observed and predicted aerosol composition averaged over each field campaign
or IOP.
• Vertical profile of cloud fraction and LWC composite of aircraft measurements for each field
campaign or IOP.
• Timeseries of atmospheric state variables.
• Aircraft and ship track maps.

In the next section we will demonstrate these diagnostics and metrics by providing several examples.

**3.  Examples**
Aerosol number concentration, size distribution, and chemical composition (that controls hygroscopicity)
are key quantities that impact aerosol-cloud interactions, such as the activation of cloud droplets. Errors in
model predictions of these aerosol properties contribute to uncertainties in aerosol direct and indirect
radiative forcing. These aerosol properties vary dramatically depending on location, altitude, season, and
meteorological conditions due to variability in emissions, formation mechanisms, and removal processes
in the atmosphere. This section shows some examples to illustrate the usage of this diagnostics package
on evaluating global models.
**3.1 Aerosol size distributions and number concentrations**
Aerosol properties are highly dependent on location and season. Figure 4 shows the mean aerosol size
distribution for each of the four testbed regions. For HI-SCALE and ACE-ENA, the two IOPs operated in
different seasons are shown separately. Table 3 shows the mean aerosol number concentration from these
field campaigns, for two particle size ranges: >10 nm and >100 nm. The 25% and 75% percentiles are
also shown to illustrate the variability in space and time. Among the four testbed regions, the CUS region
has the largest aerosol number concentrations since the other field campaigns are primarily over open
ocean. Overall, EAMv1 overestimates Aitken mode (10 – 70 nm) aerosols and underestimates
accumulation mode (70 – 400 nm) aerosols for the CUS and ENA regions, suggesting that processes
related to particle growth or coagulation might be too weak in the model. Over the NEP region, EAMv1
overestimates aerosol number for particle sizes >100 nm and >10 nm (Figure 4 and Table 3), both at the



surface and aloft. Over the SO region, which is considered a pristine region with low aerosol
concentration, observations show a significant number of particles <200 nm in both aircraft and ship
measurements. The mean aerosol number concentration over SO region is comparable or even greater
than the other ocean testbeds (Table 3). In contrast, EAMv1 simulates a clean environment with the
lowest aerosol number concentrations among the four regions. These types of comparisons demonstrate
the need for additional analyses to understand why SO has more aerosols than other ocean regions and
why EAMv1 cannot simulate this feature. The observed 75% percentiles are sometimes smaller than the
mean value (Table 3), indicating skewed aerosol size distribution with long tail in large aerosol size.
EAMv1 usually produces smaller range between 25% and 75% percentiles than the observations, likely
because the current model resolution is too coarse to capture the observed spatial variability in aerosol
properties.

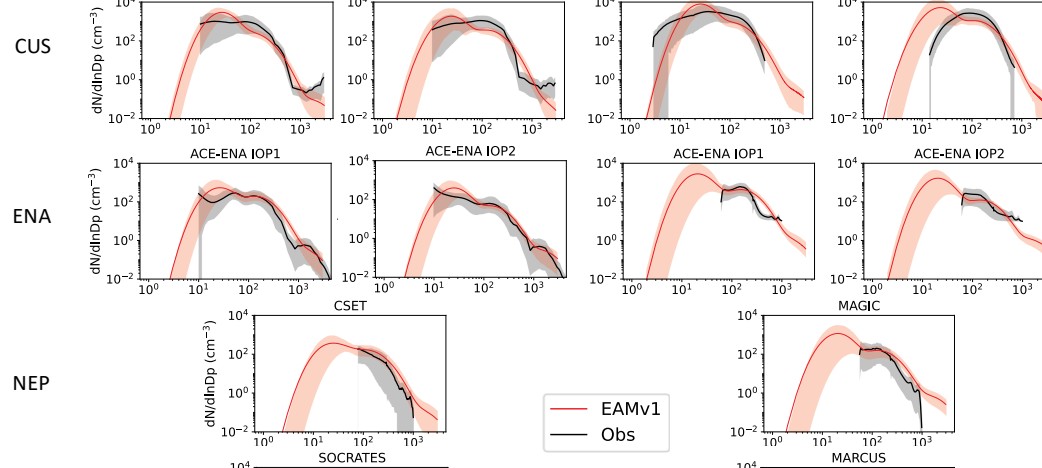


**Figure 4: Mean aerosol number distribution averaged for each field campaign or IOP. Shadings**
**denote the range between 10% and 90% percentiles.**





**Table 3: Mean aerosol number concentration and 25% and 75% percentiles (small numbers in**
**parenthesis) for two size ranges averaged for each field campaign (or each IOP for HI-SCALE and**
**ACE-ENA). Aircraft measurements 30 minutes after take-off and before landing are excluded to**
**remove possible contamination from the airport.**

| Unit: #/cm$^3$ | | | >10 nm | | >100 nm | |
|---|---|---|---|---|---|---|
| | | | CPC | E3SMv1 | UHSAS/PCASP* | E3SMv1 |
| CUS | Surface (HI-SCALE) | IOP1 | 4095 (2198, 4943) | 4566 (2865, 5984) | 675.1 (393.2, 929.5) | 321.3 (229.7, 400.8) |
| | | IOP2 | N/A | N/A | N/A | N/A |
| | Aircraft (HI-SCALE) | IOP1 | 4206 (1132, 5013) | 3872 (2803, 4946) | 465.7 (112.6, 616.1) | 159.6 (112.2, 200.5) |
| | | IOP2 | 4121 (1610, 3829) | 2514 (1332, 3584) | 789.1 (444.4, 1088.0) | 383.6 (280.7, 483.8) |
| ENA | Surface (ACE-ENA) | IOP1 | 610 (343, 711) | 1723 (600, 1650) | 206.1 (134.5, 267.1) | 209.8 (155.3, 255.5) |
| | | IOP2 | 458 (239, 505) | 843 (320, 1152) | 59.6 (25.0, 71.9) | 61.9 (53.6, 71.9) |
| | Aircraft (ACE-ENA) | IOP1 | 576 (264, 677) | 919 (562, 917) | 135.6 (65.3, 185.1) | 199.9 (146.6, 266.3) |
| | | IOP2 | 356 (132, 383) | 521 (279, 627) | 72.8 (22.2, 72.8) | 50.3 (41.6, 62.3) |
| NEP | Ship (MAGIC) | | 615 (116, 284) | 1272 (357, 1646) | 176.2 (65.3, 183.6) | 246.4 (155.9, 273.9) |
| | Aircraft (CSET) | | 408 (155, 386) | 607 (353, 675) | 81.5 (17.0, 73.4) | 134.5 (81.2, 151.3) |
| SO | Ship (MARCUS) | | 559 (270, 564) | 324 (168, 318) | 272.4 (72.5, 197.3) | 107.8 (70.7, 128.3) |
| | Aircraft (SOCRATES) | | 988 (327, 991) | 237 (169, 270) | 56.2 (14.1, 50.4) | 32.3 (13.2, 42.2) |

* PCASP is used on aircraft for HI-SCALE and ACE-ENA. UHSAS is used for others.
Both observed and simulated aerosol size distribution and number concentration show large variability
during these field campaigns. Over the period of a few weeks or longer, aerosol number can vary by an
order of magnitude between the 10% and 90% percentiles, especially for small particles (Figure 4). Figure
5 shows mean aerosol size distributions for two flight days during HI-SCALE: one with a large number of
small (<70 nm) particles (14 May) and the other (3 September) with fewer small particles but more
accumulation mode (70 – 300 nm) particles. On both days, EAMv1 reproduced the observed planetary
boundary layer (PBL) height (PBLH) reasonably well with sufficient samples below and above PBL. On



14 May, EAMv1 reproduces the observed aerosol size distribution reasonably well both within the PBL
and in the lower free atmosphere. However, on 3 September EAMv1 produces too many aerosols in the
Aitken mode and too few accumulation mode aerosols in the PBL. In the free atmosphere, EAMv1
reproduces the lower concentration of Aitken mode aerosols but still underestimates the accumulation
mode. Such contrasting cases will be useful to help diagnose the specific processes contributing to model
uncertainties in future analyses. This large day-to-day variability also indicates that long-term
measurements are needed to avoid sampling bias in building robust statistics in aerosol properties. The
next version of ESMAC Diags will be extended to include the available long-term ARM measurements at
SGP, ENA and other sites outside of the field campaign time periods.

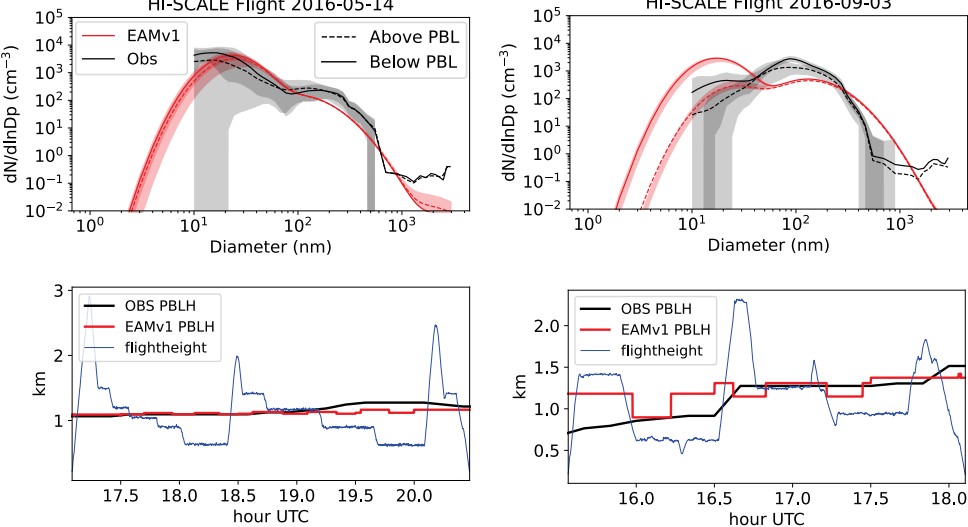


**274 Figure 5: (Top) Mean aerosol number distribution for two flights during HI-SCALE: (left) 14 May**

**275 2016 and (right) 3 September 2016, for data above (dashed line) and below (solid line) observed**

**276 PBLH. If there is cloud observed within a 1-hour window of the sample point, the above-PBL**

**277 sample needs to be above cloud top and the below-PBL sample needs to be below cloud base for the**

**278 sample point to be chosen. Shadings represent the data range between 10% and 90% percentiles.**

**279 Relatively large particles with no shading indicate more than 90% of samples with zero values.**

**280 (bottom) Timeseries of observed (black) and simulated (red) PBLH overlaid with flight height**

**281 (blue) during the two flight periods. The observed PBLH is derived from Doppler lidar**

**282 measurements.**

283



### 3.2 Vertical profiles of aerosol properties

A research aircraft is the primary platform to provide information on the vertical variations of key aerosol properties that cannot be obtained accurately by remote sensing instrumentation. In this section we show an example of evaluating vertical profiles of aerosol properties using aircraft measurements as well as illustrate the capability of evaluating multiple model simulations with ESMAC Diags. In addition to the standard EAMv1 simulation described in the previous section, we performed an EAMv1 simulation using the regionally refined mesh (RRM) (Tang et al., 2019). The model is configured to run with the horizontal grid spacing of ~0.25° over the continental U.S. and ~1° elsewhere. The two model configurations are identical, except for the higher spatial resolution (including primary aerosol emissions) in the RRM over the continental U.S. All aircraft measurements with a cloud detected simultaneously (cloud flag = 1) were excluded.

Figure 6 shows vertical percentiles of aerosol number concentration, composition and CCN number concentration among all the HI-SCALE aircraft flights. Note that aircraft rarely flew above 3 km during HI-SCALE so the sample size above that altitude is much smaller. All observed aerosol properties decrease with height since the major source of aerosols is from precursors emitted near the surface and chemical formation within the PBL. EAMv1 generally simulates less variability than observations except for sulfate. Overall, EAMv1 reproduces the observed mean aerosol number concentration for aerosol size > 10 nm but underestimates the number of larger particles > 100 nm during HI-SCALE (Table 3). The model also overestimates sulfate and underestimates organic matter concentrations when compared to aircraft AMS measurements. Its underestimation of CCN number concentration is consistent with underestimation of aerosol number concentration for diameter > 100 nm but contrary with overestimation of sulfate. A similar relationship is seen for ACE-ENA, to be described later in this section.



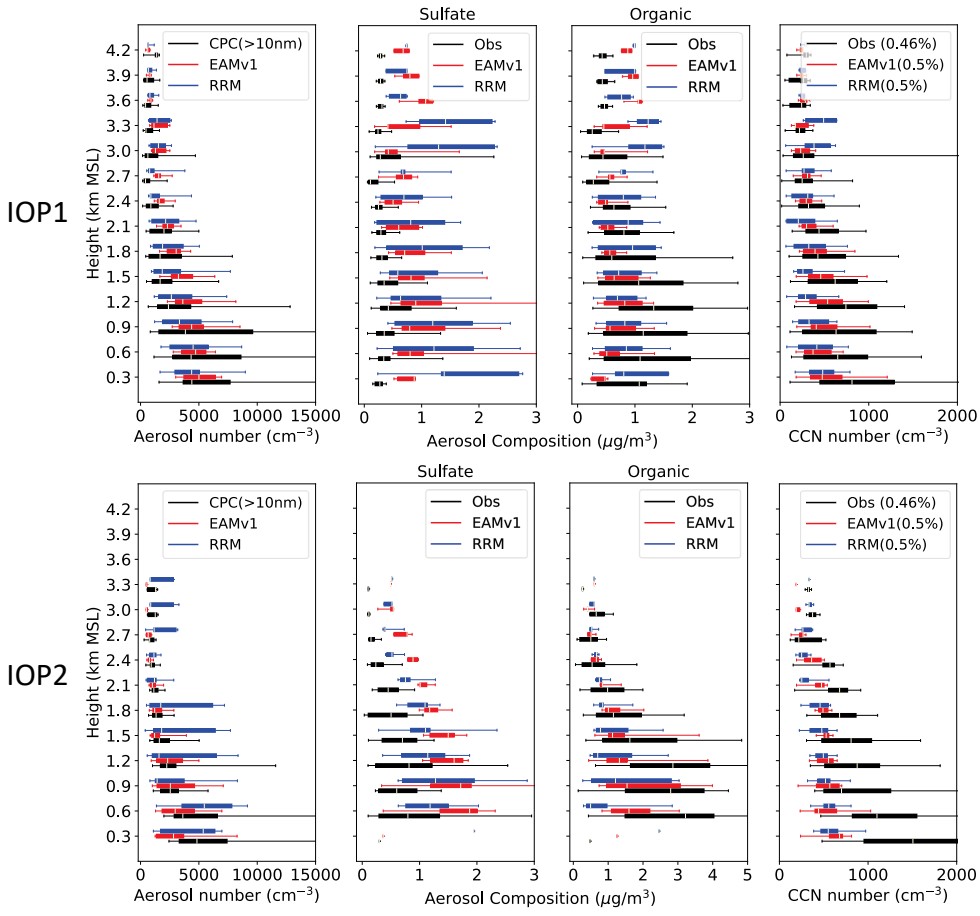

306

**Figure 6: Vertical profiles of (from left to right): aerosol number concentration, mass concentration of sulfate, mass concentration of total organic matter and CCN number concentration under the supersaturation in the parentheses for HI-SCALE (top) IOP1 and (bottom) IOP2. The percentile box represents 25% and 75% percentiles, and the bar represents 5% and 95% percentiles.**


The differences in sulfate and organic matter aloft is consistent with longer term surface measurement
differences shown in Figure 7, suggesting this is a model bias. The greater fraction of sulfate in EAMv1
suggests that the simulated aerosol hygroscopicity is likely higher than observed. Currently only these
two species are available in both EAMv1 and AMS/ACSM observations for comparison purpose. Zaveri
et al. (2021) recently added chemistry associated with $NO_3$ formation in MAM4, which is expected to be
implemented in a future version of EAM.





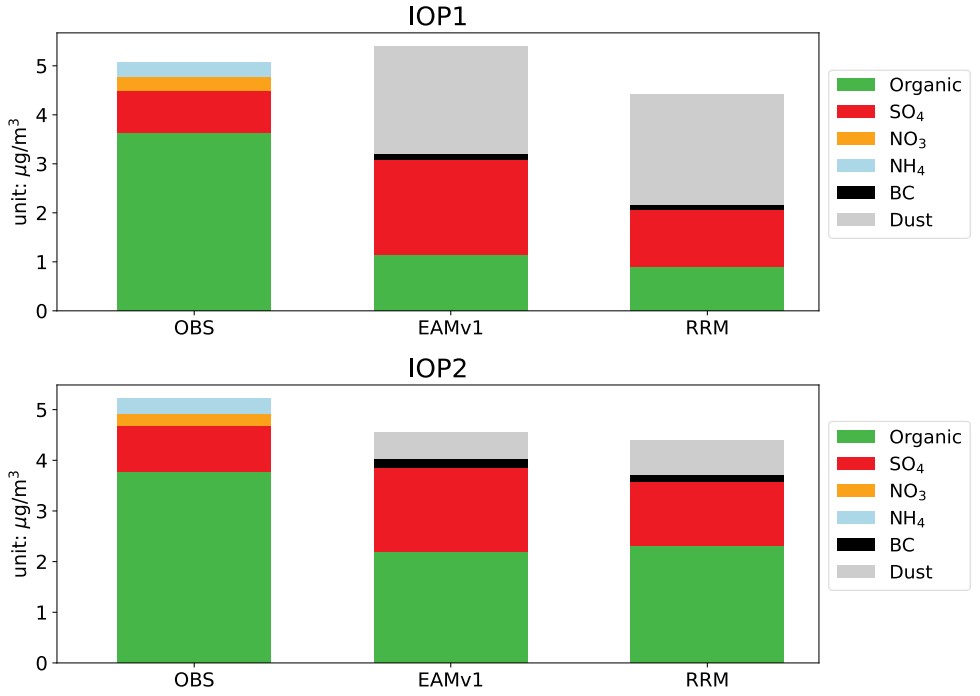

**Figure 7: Bar plots of the surface average aerosol composition during HI-SCALE IOP1 (top) and IOP2 (bottom). Observations are obtained from an ACSM. Dust and black carbon (BC) are not measured in the observation. NO₃ and NH₄ are not predicted in EAMv1 and RRM.**

Ongoing developments in E3SM will soon permit regional-refined meshes with grid spacings as small as ~ 3 km as well as global convection-permitting simulations (Δx ~ 3 km); therefore, this diagnostics package is designed to be flexible in scale to take advantage of higher-resolution ESM simulations that are more compatible with high-resolution in-situ aerosol observations. This study demonstrates this ability by using a 0.25° RRM simulation. Overall, the RRM analyzed here has similar biases as EAMv1, with differences that vary seasonally. The 25% to 75% percentiles in Figure 6 show that the variability of organic aerosols and CCN from the EAMv1 and RRM simulations are similar. However, the variability of sulfate in RRM is larger than EAMv1 and observations during the spring IOP (IOP1). During the summer IOP (IOP2), the variabilities of sulfate in EAMv1, RRM, and observations are similar, and the sulfate concentrations from RRM are closer to observed than EAMv1. Individual timeseries from the RRM simulation are still too smooth to capture the fine scale variability of aerosols in observations (not shown). We expect E3SM to capture more fine scale variabilities related to urban and point sources of aerosols and their precursors when the simulation grid spacing is further reduced to ~ 3 km. A sensitivity study will be conducted when this high-resolution version of E3SM simulation becomes available.





Figure 8 shows the vertical variation in percentiles of aerosol properties for ACE-ENA. The observed
aerosol number concentrations, composition masses, and CCN number concentrations are much smaller
than those for HI-SCALE, representing a cleaner ocean environment. EAMv1 produces larger mean
values than the observations for all these quantities. The overall variabilities in predicted aerosol number
and concentrations of sulfate and organic matter are also greater than observed. Note that the observed
variabilities for HI-SCALE are much larger than for ACE-ENA, indicating that EAMv1 has smaller
location variation on aerosol variabilities. The observed total organic concentration shows a peak aloft
between 1.6 and 2.2 km, corresponding to the level of CCN number concentration peak. This implies a
major source of aerosols or precursors is free tropospheric transport (Zawadowicz et al., 2021). This peak
of total organic concentration aloft is also captured by the model.
The bar plots in Figure 9 of aerosol composition at the surface during ACE-ENA from the ACSM
instrument and EAMv1 illustrate a similar bias in sulfate and organic mass as aloft. While the surface
sulfate measurements are like those from the aircraft at the lowest altitudes, the observed surface organic
matter is much higher than aloft, particularly during IOP2. The differences in these measurements may be
due to local effects or possible contamination from aircraft since the surface station is located near an
airport on an island.



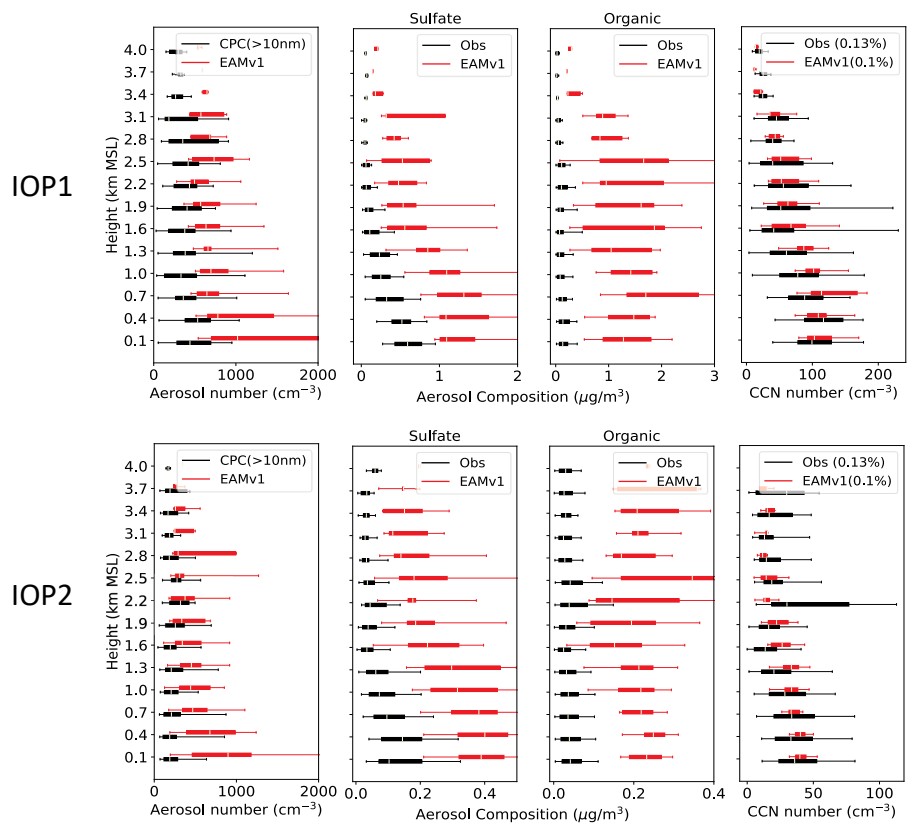


**Figure 8: Same as Figure 6 but for ACE-ENA.**

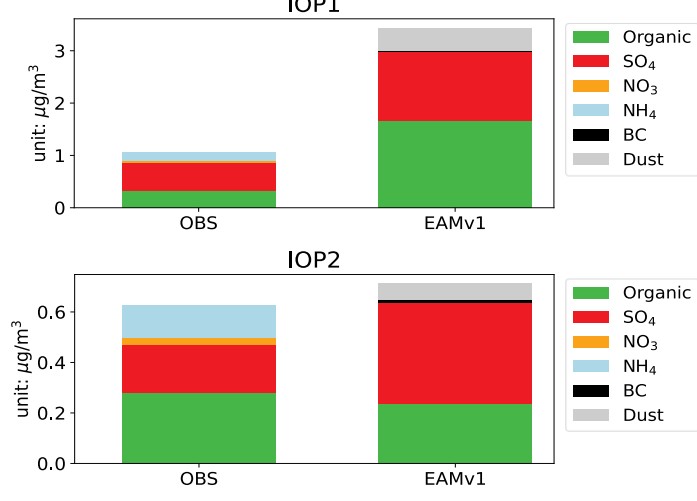


**Figure 9: Same as Figure 7 but for ACE-ENA.**

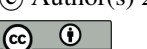

### 3.3 New particle formation events


Aerosol number concentrations and size distributions are highly impacted by new particle formation
(NPF) events (Kulmala et al., 2004), which further influence CCN concentration (e.g., Kuang et al., 2009;
Pierce and Adams, 2009) and ultimately cloud properties. NPF and subsequent particle growth are
frequently observed in the CUS region (Hodshire et al., 2016). As described by Fast et al. (2019) and
shown in Figure 10a, several NPF events were observed during the HI-SCALE spring IOP (IOP1). Large
concentrations of aerosols smaller than 10 nm were observed, with the size growing larger over the next
few hours. The average diurnal variation in aerosol number distribution in Figure 11a shows that NPF
events usually occur during the morning between 12 and 15 UTC (6 – 9 am local time), followed by
particle growth during the rest of the morning and afternoon. This variation is also seen in the diurnally
averaged CPC measurements of aerosol diameters > 3 nm and > 10 nm (Figure 11c) but diurnal changes
in CCN number concentrations (Figure 11d) are more modest.

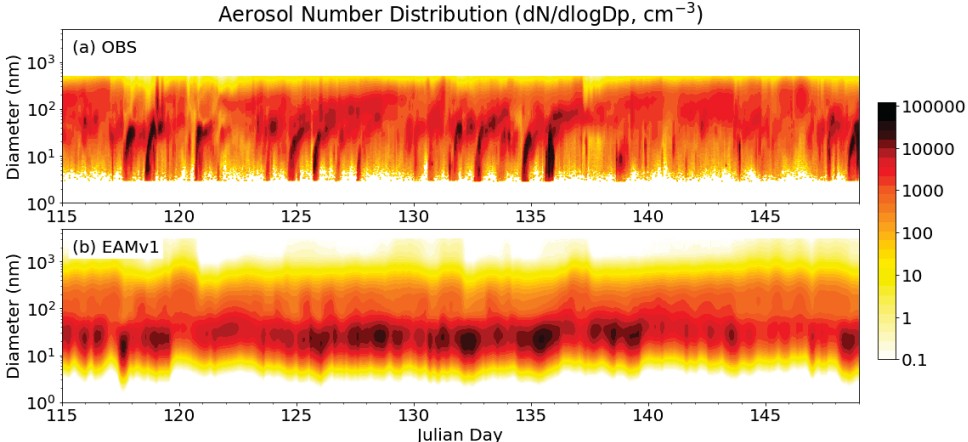


**Figure 10: Time series of (a) observed and (b) simulated surface aerosol number distribution**
**during HI-SCALE IOP1. The observed aerosol number distribution is from merged nanoSMPS**
**and SMPS.**





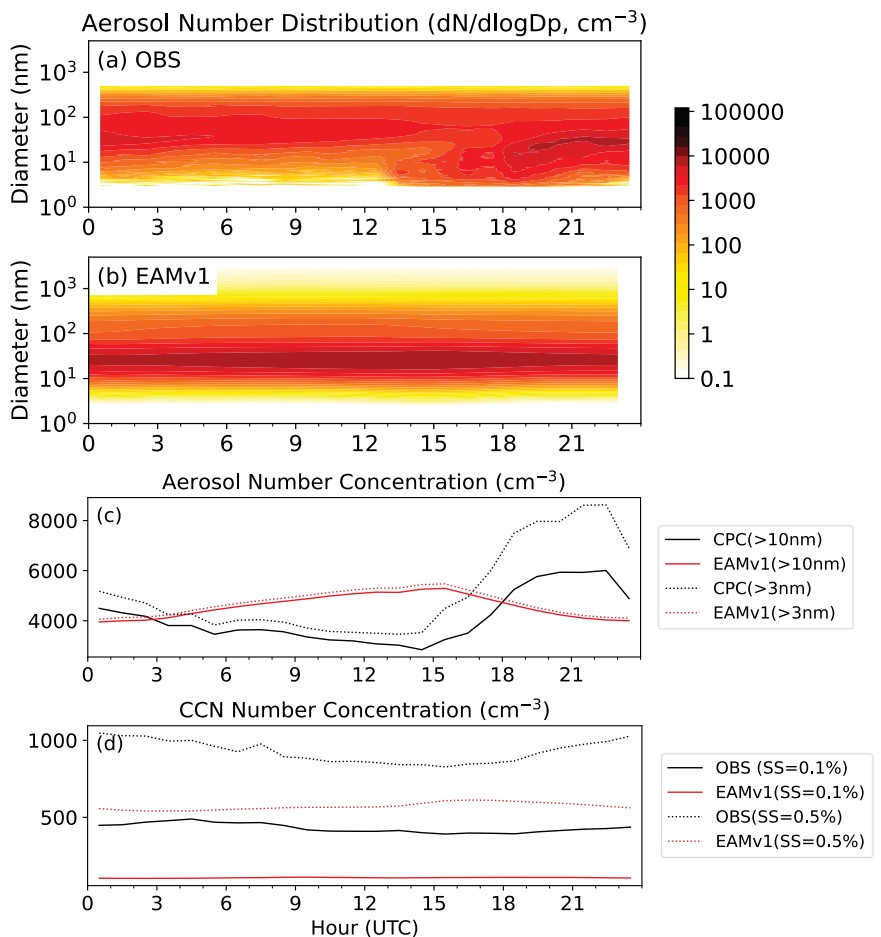

**Figure 11: Average diurnal cycle of surface (a) observed aerosol number distribution, (b) simulated aerosol number distribution, (c) aerosol number concentration for diameters > 10 nm and > 3 nm, and (d) CCN number concentration for supersaturations of 0.1% and 0.5% for HI-SCALE IOP1.**

Various NPF pathways associated with different chemical species have been proposed and implemented in models. Two NPF pathways are considered in MAM4 in EAMv1: a binary nucleation pathway and a PBL cluster nucleation pathway. However, the current simulation does not reproduce the observed large day-to-day variability of small particle concentrations due to NPF. Instead, the model produces high aerosol concentrations between 10 and 100 nm almost all the time. It also fails to reproduce the large diurnal variability of aerosol and CCN number concentration with a peak seen in the morning near 15 UTC (9 am local time), 7 hours earlier than the observed 22 UTC (4 pm local time) afternoon





peak. Its overestimation of aerosol number concentration for particle diameter >10 nm and
underestimation of CCN number concentration is consistent with that shown in Figure 4. Several efforts
are underway to improve the simulation of NPF by adding a nucleation mode in MAM4 to explicitly
resolve ultrafine particles and implementing new chemical pathways to simulate NPF following Zhao et
al. (2020). ESMAC Diags is being used to evaluate these new model developments.
Using aircraft measurements from ACE-ENA, Zheng et al. (2021) recently found evidence of NPF
events occurring in the upper part of marine boundary layer between broken clouds following the passage
of a cold front. 16 February 2018 is identified as a typical NPF day in Zheng et al. (2021). The vertical
profiles of aerosol number and CCN concentrations measured by aircraft on 16 February 2018 are shown
in Figure 12. The NPF event and particle growth happened in the upper boundary layer is shown by the
large mean and variance of aerosol number concentration just below the base of the marine boundary
layer clouds. EAMv1 could not simulate NPF events in the upper marine boundary layer on this day and
other days during ACE-ENA, likely due to the lack of NPF mechanisms related to dimethyl sulfide
oxidation, and/or missing part in parameterizations to deal with the processes related to broken marine
boundary layer clouds and sub-grid circulation. Similarly, the sharp increase of CCN number just above
the level of marine boundary layer clouds is not simulated. The differences in observed and simulated
CCN suggests that simulated aerosol-cloud interactions are not likely to be representative even though the
simulated cloud height and depth agrees reasonably well with the aircraft measurements for this day.

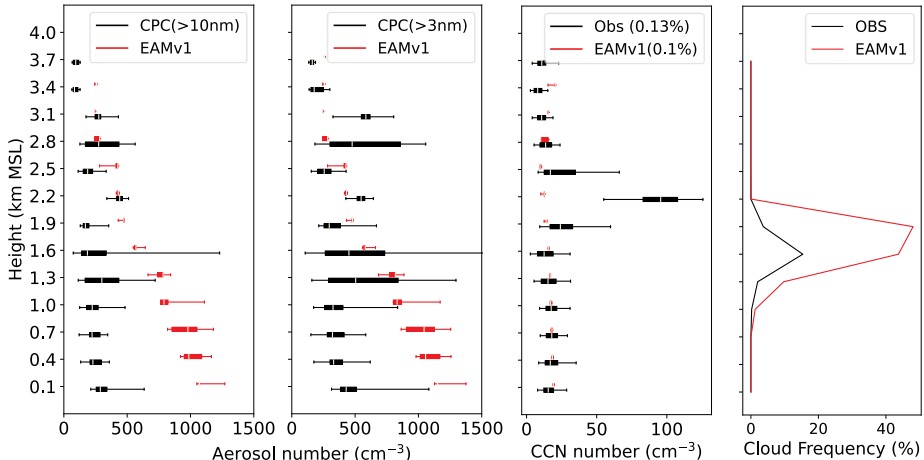


**Figure 12: Vertical profiles of aerosol number concentration for diameters >10 nm and >3 nm,**
**CCN number concentration, and cloud frequency measured by the 16 February 2018 flight in**
**ACE-ENA. The percentile box represents 25% and 75% percentiles, and the bar represents 5%**
**and 95% percentiles.**






### 3.4 Latitudinal dependence of aerosols and clouds

Unlike some field campaigns (i.e., HI-SCALE and ACE-ENA) where aircraft missions were conducted
over a relatively localized region with limited spatial variability of the meteorological conditions, ship
and/or aircraft measurements over the NEP and SO testbed regions span regions > 1500 km (i.e., from
California to Hawaii and from Tasmania to the far Southern Ocean, respectively). As shown in Figure 2,
there are large spatial gradients in EAMv1 simulated aerosol optical depth along these ship/aircraft tracks.
In ESMAC Diags version 1.0, we include composite plots of aerosol and cloud properties binned by
latitude to assess model representation of synoptic-scale variations.

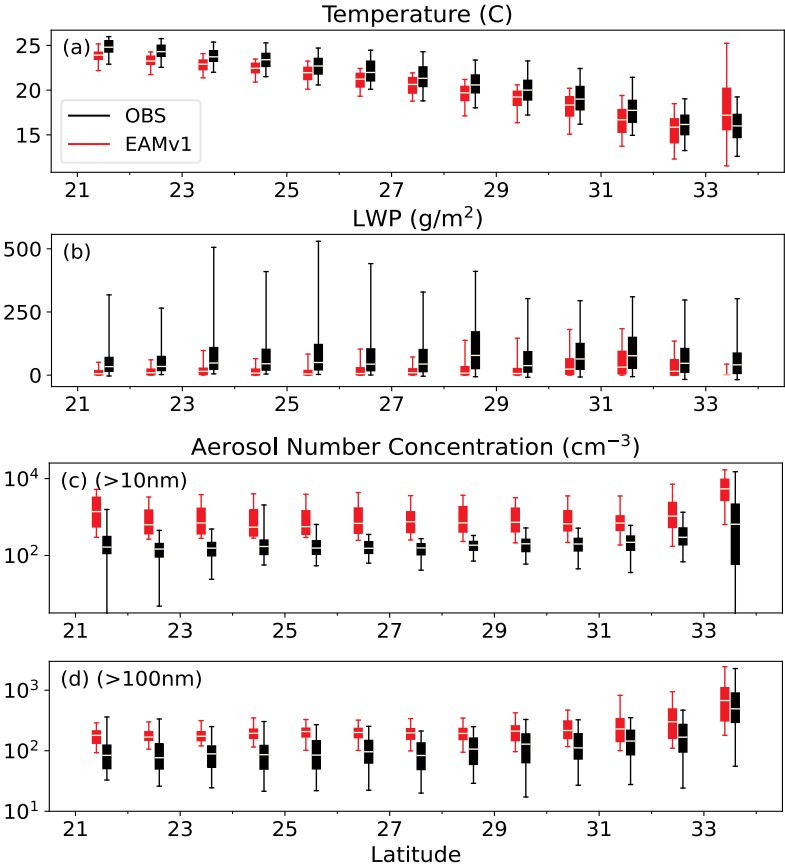


**Figure 13: Percentiles of (a) air temperature, (b) grid-mean liquid water path (LWP), (c) aerosol**

**number concentration for diameter >10 nm, and (d) aerosol number concentration for**

**diameter >100 nm for all ship tracks in MAGIC binned by 1° latitude bins. The percentile box**





**represents 25% and 75% percentiles, and the bar represents 5% and 95% percentiles. The observed aerosol number concentrations for diameters >10 nm and >100 nm are obtained from CPC and UHSAS, respectively.**

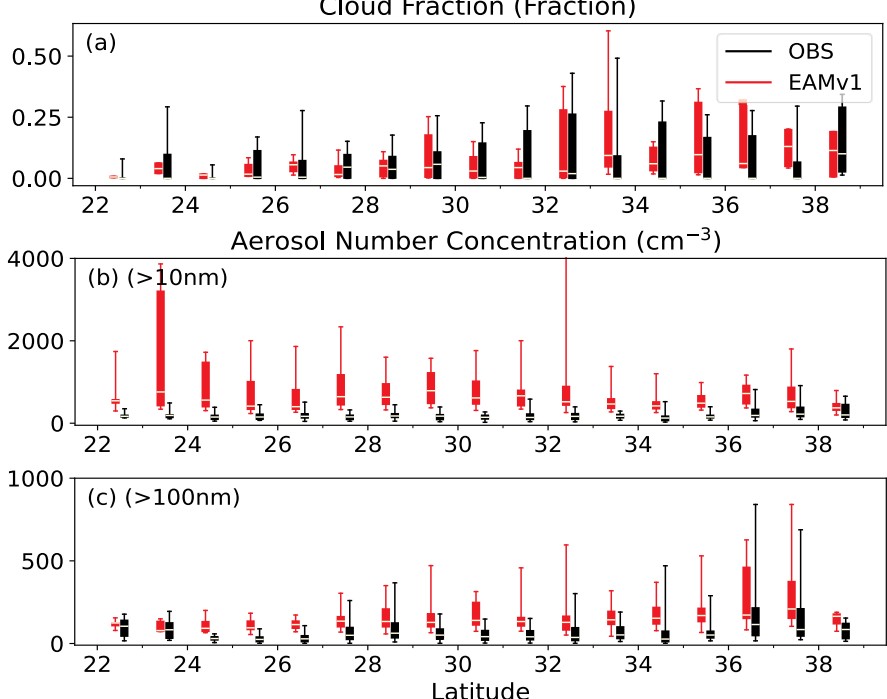

**Figure 14: Percentiles of (a) cloud fraction, (b) aerosol number concentration for diameter >10 nm, and (c) aerosol number concentration for diameter >100 nm for all aircraft measurements between 0-3 km in CSET binned by 1° latitude bins. The percentile box represents 25% and 75% percentiles, and the bar represents 5% and 95% percentiles. The observed aerosol number concentrations for diameters >10 nm and >100 nm are obtained from CNC and UHSAS, respectively.**

The research ship (aircraft) from the MAGIC (CSET) field campaign in the NEP testbed travelled between California and Hawaii, where there is frequently a transition between marine stratocumulus clouds near California and broken trade cumulus clouds near Hawaii (e.g., Teixeira et al., 2011). Some of the meteorological, cloud, and aerosol properties along the ship (aircraft) tracks binned by latitude are





shown in Figure 13 (Figure 14). Note that cloud fraction in Figure 14 is calculated as cloud frequency in
aircraft observation and from grid-mean cloud fraction in model along the flight track. This is different
from the classic definition of cloud fraction usually used for satellite measurements or models and is
subject to aircraft sampling strategy. As the surface temperature decreases from Hawaii to California
(Figure 13a), the cloud fraction (Figure 14a) shows an increasing trend, indicating the transition from
marine cumulus to stratocumulus clouds. However, ship-measured LWP (Figure 13b) has no trend related
to latitude, possibly because cumulus clouds at lower latitudes have smaller cloud fraction but larger LWP
when clouds exist. EAMv1 shows increasing trends of both cloud fraction and LWP from low to high
latitudes along these tracks. It generally underestimates LWP and overestimates cloud fraction to the
north of 30° N. Additional cloud properties derived from surface and satellite measurements are not
included in the current analysis, which was constructed to focus on aerosols. They are planned to be
included in future versions. For aerosol number concentrations, EAMv1 produces too many aerosols
compared to measurements both at the surface (ship) and aloft (aircraft), consistent with the aerosol size
distribution in Figure 4 and total number concentration in Table 3. However, EAMv1 does reproduce the
increase trend in accumulation mode aerosol concentration approaching the California coast.
Similar plots along ship tracks from MARCUS and aircraft tracks from SOCRATES are shown in
Figures 15 and 16, respectively. Over the SO region, EAMv1 simulates smaller LWP (Figure 15b) but
higher cloud fraction (Figure 16a) than observations, similar to the biases seen over the NEP region.
Aerosols measured by ship and aircraft both show large variations in number concentration at any given
latitude bin over the campaign period, while EAMv1 generally produces lower mean concentrations and
smaller variability. This indicates that the physical and chemical processes related to aerosol lifetime over
the Southern Ocean need to be better understood and represented by EAMv1.





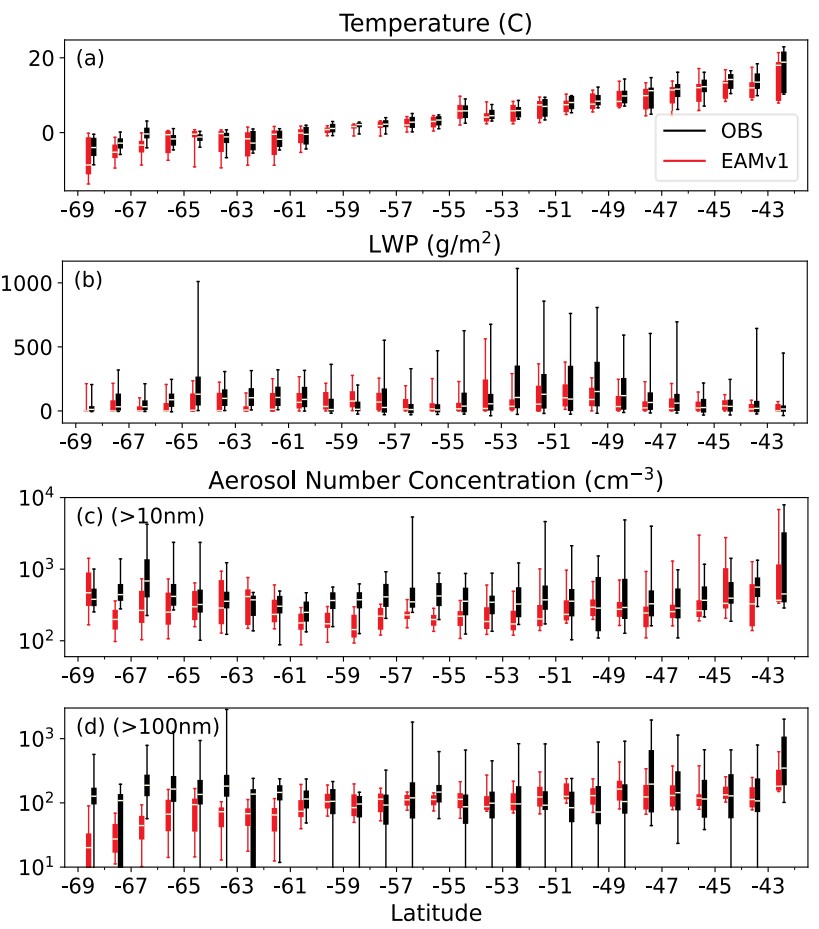


**Figure 15: Same as Figure 13 but for MARCUS.**




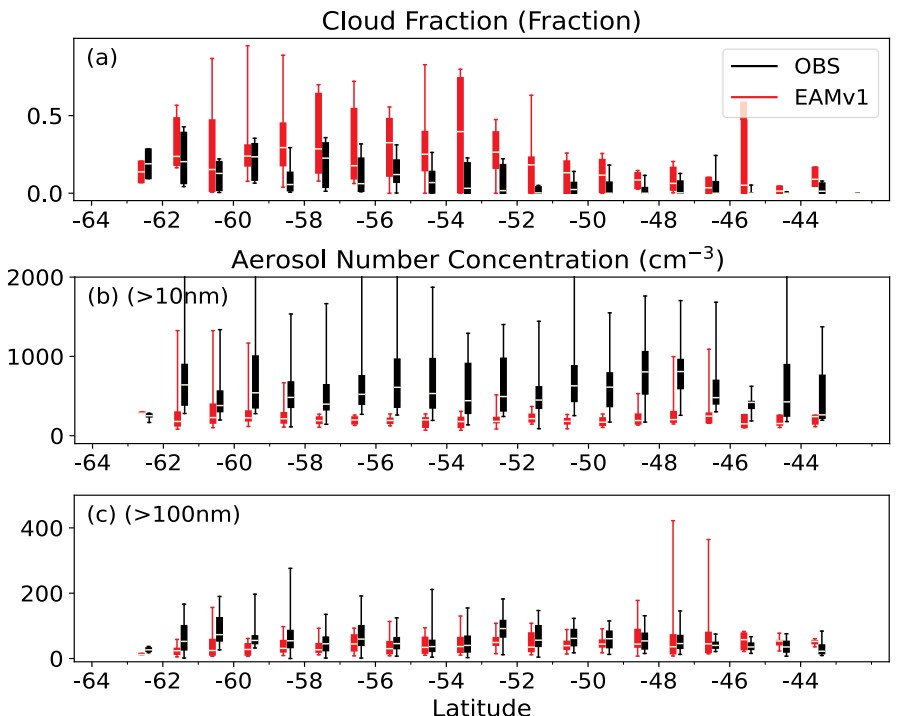


**Figure 16: Same as Figure 14 but for SOCRATES.**


### 4. Summary

A Python-based ESM aerosol-cloud diagnostics (ESMAC Diags) package is developed to quantify the performance of the DOE's E3SM atmospheric model using ARM and NCAR field campaign measurements. The first version of this diagnostics package focuses on aerosol properties. The measurements include aerosol number, size distribution, chemical composition, and CCN collected from surface, aircraft, and ship platforms needed to assess how well the aerosol lifecycle is represented across spatial and temporal scales which will subsequently impact uncertainties in aerosol radiative forcing estimates. Currently, the diagnostics cover the field campaigns of ACE-ENA, HI-SCALE, MAGIC/CSET, and MARCUS/SOCRATES over Northeastern Atlantic, Continental U.S., Northeastern Pacific, and Southern Ocean, respectively. The code structure is designed to be flexible and modular so that evaluation against new field campaigns or additional datasets can be easily implemented. Since there is no one instrument that can measure the entire aerosol size distribution, we have constructed merged aerosol size distributions from two or more ARM instruments to better assess predicted size distributions. An "aircraft simulator" is used to extract aerosol and meteorological model variables along flight paths



that vary in space and time. Similarly, the aircraft simulator is applied to ship tracks in which the altitude
remains fixed at sea level.

481       The version 1.0 of ESMAC Diags package can provide various types of diagnostics and metrics,

including timeseries, diurnal cycles, mean aerosol size distribution, pie charts for aerosol composition,
percentiles by height, percentiles by latitude, mean statistics of aerosol number concentration, and more.
A full set of diagnostics plots and metrics for simulations used in this paper are available at
[https://portal.nersc.gov/project/m3525/sqtang/ESMAC_Diags_v1/forGMD/webpage/](https://portal.nersc.gov/project/m3525/sqtang/ESMAC_Diags_v1/forGMD/webpage/). This allows
quantification of model performance predicting aerosol number, size, composition, vertical distribution,
spatial distribution (along ship tracks or aircraft tracks) and new particle formation events. This paper
shows some examples to demonstrate the capability of ESMAC Diags to evaluate EAMv1 simulated
aerosol properties. The diagnostics package also allows multiple simulations in one plot to compare
different models or model versions. It can also be applied to evaluate other ESMs with small
modifications to process model output.

492       Because in-situ aerosol measurements are usually collected at high temporal frequency (typically 1

second to a minute) over fine spatial volumes, there is a spatiotemporal scale mismatch with the standard
climate model resolution (usually 1-degree grid spacing with hourly output). This is a limitation that
cannot be completely overcome and must be accepted to perform model-observation comparisons
necessary for identifying shortcomings in model representation of aerosol, cloud, and aerosol-cloud
interaction processes that are the primary source for uncertainties in prediction of future climate. As new
versions of E3SM become available that has grid spacings as small as a few kilometers via regional-
refined and convection-permitting global domains (e.g., Caldwell et al., 2021), spatiotemporal
variabilities of aerosols at finer scales should be captured and be more compatible with fine resolution
observations such that resolution impacts on statistical differences can be quantified. The diagnostics
package will be applied to diagnose high resolution model output when the data are available.

503       While the current version focuses on aerosol properties, a version 2 of ESMAC Diags is being planned

to include more diagnostics and metrics for cloud, precipitation, and radiation properties to facilitate the
evaluation of aerosol-cloud interactions. These include inversion strength, above cloud relative humidity,
cloud-surface coupling, cloud fraction, depth, LWP, optical depth, effective radius, droplet number
concentration, adiabaticity, and albedo, precipitation rate, and more. Analyses are being designed to
quantify relationships between these variables and relate them to effective radiative forcing, which will be
used to assess and improve model parameterizations. Additional surface-based and satellite retrievals will
also be used to address limitations related to data coverage and uncertainty. While there are other efforts
to develop model diagnostics packages, this diagnostics package provides a unique capability for detailed



evaluation of aerosol properties that are tightly connected with parameterized processes. Together with
other commonly used diagnostics packages such as the ARM diagnostics package (Zhang et al., 2020),
the DOE E3SM diagnostics package, and the PCMDI's metrics package (Gleckler et al., 2016), we expect
to better understand the strengths and weaknesses of E3SM or other ESMs and provide insights into
model deficiencies to guide future model development. This includes studies that develop a better
understanding of how various processes contribute to uncertainties in aerosol number and composition
predictions and subsequent representation of CCN and aerosol radiative forcing estimates.



**Code availability**:

*The current version of ESMAC Diags is publicly available through GitHub (https://github.com/eagles-project/ESMAC_diags) under the new BSD license. The exact version (1.0.0-alpha) of the model used to produce the results used in this paper is archived on Zenodo (https://doi.org/10.5281/zenodo.5733233).*

**Data availability**:

*Measurements from the HI-SCALE, ACE-ENA, MAGIC, and MARCUS campaigns as well as the SGP and ENA sites are supported by the DOE Atmospheric Radiation Measurement (ARM) user facility and available at https://adc.arm.gov/discovery/. Measurements from the CSET and SOCRATES campaigns are supported by National Science Foundation (NSF) and obtained from NCAR Earth Observing Laboratory at https://data.eol.ucar.edu/master_lists/generated/cset/ and https://data.eol.ucar.edu/master_lists/generated/socrates/, respectively. DOI numbers or references of individual instruments are given in Table 2. All the above observational data and preprocessed model data used to produce the results used in this paper is archived on Zenodo (https://doi.org/10.5281/zenodo.5669136).*

**Author contribution**:

*ST, JDF and PM designed the diagnostics package; ST wrote the code and performed the analysis; JES, FM and MAZ processed the field campaign data; KZ contributed to the model simulation; JCH and ACV contributed to the package design and setup; ST wrote the original manuscript; all authors reviewed and edited the manuscript.*

**Competing interests**:

*Po-Lun Ma is a Topical Editor of Geoscientific Model Development. Other authors declare that they have no conflict of interest.*

*Acknowledgements:*

*This study was supported by the Enabling Aerosol-cloud interactions at GLobal convection-permitting scalES (EAGLES) project (74358), funded by the U.S. Department of Energy, Office of Science, Office of Biological and Environmental Research, Earth System Model Development (ESMD) program area. We thank the numerous instrument mentors for providing the data. This research used resources of the National Energy Research Scientific Computing Center (NERSC), a U.S. Department of Energy Office of Science User Facility operated under Contract No. DE-AC02-05CH11231. Pacific Northwest National Laboratory (PNNL) is operated for DOE by Battelle Memorial Institute under contract DE-AC05-76RL01830.*



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
