# Peer review of "Earth System Model Aerosol-Cloud Diagnostics Package (ESMAC Diags) Version 1: Assessing E3SM Aerosol Predictions Using Aircraft, Ship, and Surface Measurements"

_Geoscientific Model Development, 2021_

## Author Comment (AC1)

**Reviewer 1:**

**General Comments**

The overall quality of the manuscript is in my opinion very good, and a perfect match for this journal. The writing is concise yet detailed, the displayed graphs are of high quality and informative, and the structure is easy to follow. The topic is highly relevant for the audience of GMD and the authors seem to have a great expertise in it. The proposed model evaluation software is a valuable contribution to the field of atmospheric science, deepening the impact of the incorporated measurement campaigns in the modelling community. This package could become an indispensable tool to improve aerosol-cloud interactions, particle formation and other parametrizations in the E3SM – and potentially other GCM's as well. Investments in well-documented, open-source model validation software should be encouraged because they allow the community to get the most out of the available observation data.

We would like to thank Gijs van den Oord for taking the time to review this paper and provide helpful comments to improve the paper. The comments are repeated below in black with our reply in blue.

**Specific Comments**

- Figure 1 caption: I wouldn't call this a workflow but a directory structure. I personally think a workflow – boxes representing functions and arrows representing data flows – is more informative to understand the processing steps, so I would recommend to make such a graph, and perhaps move the current directory structure of Fig. 1 to an appendix

Thank you for the comment. We added a new graphical representation of the workflow as Figure 1, and moved the directory structure diagram to Figure 2. We are keeping the directory structure diagram in the main text as it contains more information than the workflow. The text has been modified to reflect the additional figure 1 as follows:

"*The workflow of ESMAC Diags v1 is illustrated in Figure 1. In some field campaigns, more than one instrument is used to measure aerosol size distribution over different size ranges. We therefore merge these datasets to create a more complete description of the size distribution. Other field campaign datasets are directly read by the diagnostics package. These data are introduced in Section 2.1. Model outputs are extracted at the ground sites and along the flight tracks or ship tracks. The simulation and preprocessing details are provided in Section 2.2. ESMAC Diags reads in these field campaign and model data with quality controls and generates a set of diagnostics and metrics listed in Section 2.3. The diagnostics package is designed to be flexible so that additional measurements and functionality can be included in the future. Figure 2 depicts the directory structure to illustrate the organization of the datasets and code. It is relatively straightforward to add other field campaigns or datasets using this structure.*"

[Figure]

**Figure 1: Workflow of ESMAC Diags. Data preprocessing and input are indicated by blue; diagnostics and plotting are indicated by orange.**

- Line 158: Here I believe it is appropriate to actually mention the applied thresholds in the text.

We added the applied threshold in the text:

*"(e.g., 500 cm$^{-3}$ maximum threshold is used for each UHSAS bin from the NCAR research flight measurements)"*

- Line 186: This information to me seems crucial for applicability of ESMAC Diags beyond E3SM, and I would therefore clearly state what the package exactly needs from E3SM, on which resolution and which frequency, maybe even in a small table. Also it would be great to have a remark on the applicability of this software to CMIP6 data. On line 490 we again encounter a short statement about generalization beyond E3SM, and also there I believe the paper would benefit from elaborations on the necessary model output for this.

Thank you for the suggestion. We have added an appendix (also attached below) to show the namelist of E3SM hourly output so that users can apply it in their own E3SM simulations (or output similar variables if running other models) to use this package. Models beyond E3SM can be applied to ESMAC Diags v1 when they have high-frequency output containing variables in Appendix A.
The current ESMAC Diags does not support evaluating CMIP6 data, because CMIP6 data do not save hourly data (and only a few variables have 3-hourly output) and do not reproduce specific observed

events as they are not nudging towards observations. CMIP output will be more compatible with longer term surface-based and satellite observations that will be added in future versions of ESMAC Diags.

Appendix A: Namelist containing the variables and regions of E3SM hourly output over the six field campaigns used in the E3SM run script in this study. Here *fincl4* defines output variables with the 4[th] frequency (1 hr) and interval (24 per day) in *nhtfrq* and *mfilt*, respectively. *fincl4latlon* defines the latitude and longitude range of *fincl4* output.

```
nhtfrq     = 0,-24,-3,-1
mfilt      = 1,1,8,24
…
fincl4     = 'PS',      !! dynamical fields
             'U',       !! ..
             'V',       !! ..
             'T',       !! ..
             'Q',       !! vapor (kg/kg)
             'CLDLIQ',  !! cloud hydrometeors (kg/kg)
             'CLDICE',  !! ..
             'CLDTOT',
             'NUMLIQ',  !! ..
             'NUMICE',  !! ..
             'PBLH',    !! PBL height
             'LHFLX',   !! energy fluxes
             'SHFLX',   !! ..
             'FLNT',    !! ..
             'FSNT',    !! ..
             'FLNS',    !! ..
             'FSNS',    !! ..
             'TREFHT',  !! ..
             'Z3',      !! geopotential height
             'RELHUM',  !! relative humidity (RH)
             'RHW',     !! RH with respect to water
             'RHI',     !! RH with respect to ice
             'CLOUD',   !! cloud fraction
             'AWNI',    !! in-cloud values
             'AWNC',    !! Average cloud water number conc (1/m3)
             'CCN1',    !! CCN concentration at S=0.02% (#/cm3)
             'CCN3',    !! CCN concentration at S=0.1% (#/cm3)
             'CCN4',    !! CCN concentration at S=0.2% (#/cm3)
             'CCN5',    !! CCN concentration at S=0.5% (#/cm3)
             'AREI',    !! ..
             'AREL',    !! ..
             'PRECT',   !! precipitation
             'PRECC',   !! ..
             'PRECL',   !! ..
```

```
    'FICE',      !! ice mass fraction
    'IWC',       !! grid box average ice water content (kg/m3)
    'LWC',       !! grid box average liquid water content (kg/m3)
    'TGCLDLWP',  !! liquid water path (including convective clouds)
    'TGCLDIWP',  !! ice water path (including convective clouds)
    'AODVIS',    !! AOD
    'DMS',       !!
    'SO2',       !!
    'H2SO4',     !!
    'bc_a1',     !! aerosols mass (kg/kg)
    'bc_a3',     !!
    'bc_a4',     !!
    'dst_a1',    !!
    'dst_a3',    !!
    'mom_a1',    !!
    'mom_a2',    !!
    'mom_a3',    !!
    'mom_a4',    !!
    'ncl_a1',    !!
    'ncl_a2',    !!
    'ncl_a3',    !!
    'pom_a1',    !!
    'pom_a3',    !!
    'pom_a4',    !!
    'so4_a1',    !!
    'so4_a2',    !!
    'so4_a3',    !!
    'soa_a1',    !!
    'soa_a2',    !!
    'soa_a3',    !!
    'num_a1',    !! aerosols number (#/kg)
    'num_a2',    !!
    'num_a3',    !!
    'num_a4',    !!
    'num_c1',    !! aerosols number (#/kg)
    'num_c2',    !!
    'num_c3',    !!
    'num_c4',    !!
    'dgnd_a01',  !! dry aerosol size
    'dgnd_a02',  !! ..
    'dgnd_a03',  !! ..
    'dgnd_a04',  !! ..
    'dgnw_a01',  !! wet aerosol size
    'dgnw_a02',  !! ..
```

```
        'dgnw_a03',  !! ..
        'dgnw_a04',  !! ..
        'EXTINCT',   !! Aerosol extinction (1/m)
        'AODABS',    !! Aerosol absorption optical depth 550 nm
        'ABSORB',    !! Aerosol absorption (1/m)
fincl4lonlat = '260e:265e_34n:39n',  ! SGP (~5x5 degs)
        '330e:335e_37n:42n',  ! ENA
        '202e:240e_19n:40n',  ! CSET
        '202e:243e_20n:35n',  ! MAGIC
        '60e:160e_42s:70s',   ! MARCUS
        '133e:164e_42s:63s',  ! SOCRATES
```

- Line 312: I see a discrepancy between organic aerosol composition during IOP1 at 300 m height (from Fig. 6) and the surface measurements (Fig. 7); where the simulations agree with the former, the difference with the latter is striking when one looks at Fig. 7. The authors have a similar observation for the ACE-ENA campaign and address this on line 349, could that explanation cover the HI-SCALE case too?

In Fig. 6 the comparison is along the flight tracks which can be a few hundred kilometers away from the ARM site, where data in Fig. 7 is measured. The differences between ground measurements and near-surface aircraft measurements are mainly due to spatial variability of aerosol composition. We made a comparison between surface ACSM data and lower-level aircraft AMS measurements when the aircraft was flying within a few kilometers of the ARM site and found that they were consistent. We added the following sentence to explain this discrepancy:

*"Note that near-surface measurements by aircraft are not always consistent with ground measurements (e.g., total organic matter in IOP1), which reflects the large spatial variability in aerosol properties associated with the aircraft flight paths up to a few hundred kilometers around the ARM site."*

- Figure 10: The clipping of the heat map at (I believe) 700 nm due to the range of the (nano)SMPS is somewhat confusing in a comparison graph: maybe the model graph could be cut off there too? Or just limit both y-axes to that threshold?

We revised Figures 11 and 12 to apply the same cut off from the observations to the model, so that it is easier to visually compare the two panels.

- Line 390-407: This is an interesting section showcasing the ability to focus upon single events and assess the representation of aerosol-cloud interactions on shorter time scales. Is this event automatically chosen by the package, or does the user need to select this particular day by hand? Are there other interesting events the authors could mention (possibly involving precipitation)?

This case is chosen from Zheng et al. (2021). ESMAC Diags does not have the capability to choose a case automatically. We manually select this case to demonstrate that ESMAC Diags can be used to analyze individual NPF events. There are several other interesting events given in supplementary information in Zheng et al. (2021).

- Figure 14+15: It is somewhat confusing to me the authors chose to display the aerosol number concentrations for the ship measurements on a log scale and for the aircraft measurements on a linear scale.

We revised the figures for MAGIC and MARCUS to display aerosol number concentrations for both ship and aircraft measurements using a linear scale and applied this change to ESMAC Diags.

- Line 433+446: This section contains a digression into cloud scheme assessment. I understand from the summary that the authors intend to expand this capability of the package, but I would consider dropping this paragraph or moving it elsewhere because it may distract from the main topic.

We have removed the plots and the discussion on clouds over the Southern Ocean (Figures 16 and 17). However, we feel some basic meteorological and cloud fields (cloud fraction, LWP) are important over the Northeast Pacific to illustrate the transition of cloud regimes, and these comparisons are included in ESMAC Diags v1. Therefore, we decided to keep this paragraph but added the following statement:

*"Although ESMAC Diags v1 focuses primarily on aerosols, we show some basic meteorological and cloud fields here since they are important to illustrate the transition of cloud regimes along the ship (aircraft) tracks. Additional cloud properties derived from surface and satellite measurements are not included in the current analysis, but are being implemented in ESMAC Diags v2."*

- Summary section: The authors present an outlook into future development of the package, including more cloud-related diagnostics and supporting high-resolution versions of the model. Here I would expect a few sentences about **which other measurement campaigns the authors wish to include in a future version of ESMAC Diags** (or if none: why current observation datasets provide a complete assessment of aerosol processes).

The ongoing version 2 of ESMAC Diags is focusing on clouds and aerosol-cloud interactions for the field campaigns currently used in the four testbed regions. In the future, we are considering how to extend this package to other campaigns or other ESMs. We added the following statement in the summary:

*"In the future, this diagnostics package may also be extended to include other field campaigns that provide valuable data on aerosol properties and cloud-aerosol interactions, such as the ARM Layered Atlantic Smoke Interactions with Clouds (LASIC, Zuidema et al., 2018), NASA ObseRvations of Aerosols above CLouds and their intEractionS (ORACLES, Redemann et al., 2021), or NASA Atmospheric Tomography Mission (ATom, Brock et al., 2019) campaigns. As an open-source package, ESMAC Diags can also be applied by any user to other ESMs with small modifications on model preprocessing."*

**Technical Corrections**

- Line 98: SOA and MOA should be spelled out, they are mentioned first here

The full names of SOA and MOA are added:

*secondary organic aerosol (SOA), marine organic aerosol (MOA)*

- Line 154: CPC should be spelled out, it is mentioned first here

It is now spelled out: *condensation particle counter (CPC)*

---

## Author Comment (AC2)

**Reviewer 2:**

This manuscript describes the ESMAC Diags version 1.0 package and provides useful examples of its application. I have only minor concerns and some hopefully useful suggestions below, but otherwise I believe the manuscript is ready for prompt publication. Kudos to the authors for this nice service to the community, and I hope the package gets good use. -MD

We would like to thank Michael Diamond for taking the time to review this paper and provide helpful comments to improve the paper. The comments are repeated below in black with our reply in blue.

**General comments:**

**Addition of future campaigns:**

Would it be possible to discuss more about **which other campaigns are being considered for inclusion in future versions of the diagnostics package**? The southeast Atlantic smoke-cloud campaigns (NASA ORACLES, DOE LASIC, plus CLARIFY and AEROCLO-SA internationally) in particular could be great testbeds for aerosol representation and have good ground- and air-based sampling. ATom could also be really interesting for its global reach.

Although the ongoing version 2 of ESMAC Diags is focusing on clouds and aerosol-cloud interactions for current field campaigns, this package can also be extended to other campaigns or other ESMs in the future. We added the following statement in the summary:

"In the future, this diagnostics package may also be extended to include other field campaigns that provide valuable data on aerosol properties and cloud-aerosol interactions, such as the ARM Layered Atlantic Smoke Interactions with Clouds (LASIC, Zuidema et al., 2018), NASA ObseRvations of Aerosols above CLouds and their intEractionS (ORACLES, Redemann et al., 2021), or NASA Atmospheric Tomography Mission (ATom, Brock et al., 2019) campaigns. As an open-source package, ESMAC Diags can also be applied by any user to other ESMs with small modifications on model preprocessing."

**Treatment of observations as "truth":**

At some locations in the text (e.g., "underestimation" in Line 303) the language sounds like the observations are being treated as base "truth." Other locations more thoroughly discuss limitations in the observed data as well. It might be helpful to address the issue of how observations are treated (not truth, but useful baseline given limitations are known) more in the introduction or methods sections. We agree that observations have their own limitations and uncertainties although they are usually treated as "truth" when evaluating models. As suggested, we added the following discussion in Section 2.1:

"Although these measurements are considered as "truth" when evaluating ESMs, we note that they are subject to limitations and uncertainties due to theoretical/methodological formulations, sampling representativeness, instrumental accuracy and precision, imperfect calibration, random errors, etc. In addition, sampling volumes differ between observations and model output and are not reconcilable. It is difficult to quantify every aspect of observational uncertainty within the context of interpreting comparisons with model output, but we try to discuss some of them in this study to the best of our knowledge. Percentiles (either 25% - 75% or 5% - 95%) are used in some analyses of this study to approximate data variability that is likely to be much higher than measurement uncertainty."

**Specific comments:**

Line 164: Specify that size is referring to **aerodynamic dry diameter** (or whatever it is you are using) for all uses thereafter. I'm assuming diameter but I don't remember seeing it in the text (but it is in some figure labels).

We added the following sentence to specify what "size" means:

"In ESMAC Diags v1, aerosol "size" refers to mobility and optical dry diameter of particles."

Line 208: Why choose only latitude? Are there any longitudinal variation issues that should be addressed?

The purpose of this bullet of diagnostics is examining variations due to climate regime transitions along aircraft or ship tracks across the Northeastern Pacific or the Southern Ocean. We chose latitude in this diagnostics package because the latitudinal gradient dominates over the Southern Ocean, while over the Northeastern Pacific variations exist both longitudinally and latitudinally. We can add similar diagnostics along longitude if additional field campaigns are incorporated where longitudinal variation is relevant.

Figures 4-5: It might be helpful to also place markers binned for >10 nm and >100 nm for easy comparison to Table 3 and the discussion in the text.

The figures are updated with grid lines to easily check 10 nm and 100 nm bins, and this modification has been made to ESMAC Diags output.

Line 256 (Table 3): Are there any issues worth discussing between the PCASP and UHSAS data, e.g., different size cutoffs and bins?

It is a data availability issue that for aircraft measurements during HI-SCALE and ACE-ENA, only PCASP is available. UHSAS is available on surface measurements during HI-SCALE and ACE-ENA, and in other field campaigns. Size cuts of the respective size distribution measurements are given in Table 2. We revised the sentence as below to avoid confusion:

"PCASP is available only on aircraft for HI-SCALE and ACE-ENA. UHSAS is available only in surface measurements for HI-SCALE and ACE-ENA, and in other field campaigns."

Lines 297-299: A citation from a relevant HI-SCALE paper would be useful here.

We revised this sentence and added a citation as below:

"All observed aerosol properties decrease with height since the major sources of aerosols (anthropogenic, biogenic, and biomass burning) (Liu et al., 2021) are from precursors emitted near the surface and chemical formation within the PBL."

*Reference: Liu, J., Alexander, L., Fast, J. D., Lindenmaier, R., and Shilling, J. E.: Aerosol characteristics at the Southern Great Plains site during the HI-SCALE campaign, Atmos. Chem. Phys., 21, 5101-5116, https://doi.org/10.5194/acp-21-5101-2021, 2021.*

Lines 400-401: I'm not sure this concern is warranted, as above-cloud CCN concentration has only limited relevance to cloud properties because the timescale for entraining above-cloud air into the cloudy boundary layer is on the order of days (Diamond et al., 2018; Mardi et al., 2019). The below-cloud CCN concentrations seems better-represented, and these should be the relevant metric for ACI considerations.

**We agree with the reviewer and delete this statement.**

Lines 440-441: Although this reads the "right" way based on the x-axis in Figures 13-14(a), it's backwards from the Lagrangian/cloud perspective. I'd recommend flipping it ("SSTs increase from CA to HI...").

**We revised the corresponding statements to flip all description from CA to HI.**

Lines 452-458: The commentary here is really sparse as compared to the other sections/campaigns. I'm not as familiar with this region, but I know of a few papers from Daniel and Isabel McCoy and colleagues that seem potentially relevant (listed below), and am sure there are many others that could be usefully discussed here.

Thank you for the suggestion. We have now expanded the discussions in Southern Ocean below:

"Similar latitudinal gradients of aerosol and CCN number concentrations along ship tracks from MARCUS and aircraft tracks from SOCRATES are shown in Figures 16 and 17, respectively. Over the SO region, NPF frequently occurs during austral summer when ample biogenic precursor gases (e.g., DMS) are released and rise into the free troposphere (McFarquhar et al., 2021; McCoy et al., 2021). Large values of shipmeasured aerosol and CCN number concentration are observed near Antarctica corresponding to the coastal biological emissions of aerosol precursors, and also occur to the north of 45°S, indicating impacts from continental and anthropogenic sources. This is consistent with other studies (Sanchez et al., 2021; Humphries et al., 2021). EAMv1 underestimates aerosol and CCN number concentration near Antarctica. This bias, which may be related to too strong wet scavenging or insufficient NPF and growth, is commonly seen in many other ESMs (e.g., McCoy et al., 2020; McCoy et al., 2021). Aircraft flight paths during SOCRATES (Figure 17) do not extent as far south as the ship measurements (Figure 16). The observed aerosol properties have little latitudinal variation in general. EAMv1 underestimates aerosol number concentration for size > 100m and CCN number concentration with SS=0.5%, but the predictions are closer to observed for aerosol size > 100 nm and CCN with SS=0.1% (Figure 17), consistent with the mean aerosol size distribution in Figure 5. This indicates that the model performs better in simulating accumulation mode than Aitken mode particles. These model aerosol biases are highly relevant when considering their interaction with clouds and radiations, which will be included in version 2 of ESMAC Diags."

---

## Author Response (AR2)

Dear Editor(s),
Thank you for reviewing the manuscript. We have addressed all the raised points and
revised the manuscript accordingly. Please see below for the detailed answer (in blue)
for each point.
Best regards,
Shuaiqi Tang

Dear Authors,

Thank you for revising the manuscript. I am providing below a set of remarks based on
reading the revised version of the paper and the reviews.

Please address the following points:

* Appendix A requires a substantial refactoring if aimed for publication both in terms
of content and presentation. Units and descriptions should be provided for all items,
and all rows must have understandable meaning. NB, "namelist" is not a part of Python
nomenclature or jargon.

*As response to a later comment, we uploaded the model configuration and execution scripts as electronic
supplement, which include the content in Appendix A. Therefore, we removed this appendix from the
main text.*

* In the reply to comments from both reviewers, there is a mention of newly added "In
the future, this diagnostics package ..." paragraph with new references (Zuidema et al.
2018, Redemann et al., 2021, Brock et al., 2019), which cannot be found in the
uploaded revised manuscript.

*Thank you for catching this error. This paragraph is now added in the revised manuscript (line 559-565 in
the revised manuscript).*

* Reproducibility: please add information to the Code Availability section on the way to
and any legal constraints to obtain code and reproduce results of the EAMv1/MAM4
model simulations ("In this study, we run the model from 2012 to 2018", ... "we
performed an EAMv1 simulation using the regionally refined mesh ..."). Note that
according to GMD guidelines, such specification includes precise version specification
of the model code as well as configuration files, model execution automation scripts as

well as output analysis scripts. All these files must be stored in a persistent repository such as Zenodo (or provided as an electronic supplement to the paper).

*We added version information and DOI for the model code used in this study:*

*"The model simulation used in this paper is version 1.0 of E3SM (https://doi.org/10.11578/E3SM/dc.20180418.36). The model configuration and execution scripts are uploaded as electronic supplement to this paper."*

*The model configuration and execution scripts are uploaded as electronic supplement of this paper.*

\* page 30 / line 535 - please archive this resource as well (can be either Zenodo or electronic supplement to the paper)
*we archived this resource as an electronic supplement of the paper and revised the text regarding this supplement:*

*"A full set of diagnostics plots and metrics for simulations used in this paper are available at https://portal.nersc.gov/project/m3525/sqtang/ESMAC_Diags_v1/forGMD/webpage/ and archived as an electronic supplement of this paper."*

\* Similarly as with reviewer-suggested specification of what "size" means (mobility/optical dry diameter), I suggest to specify if concentrations are given "at standard temperature and pressure" conditions or in ambient conditions or where which option is used. (see, e.g., https://scholar.google.com/scholar?q=+concentration+"standard+temperature+and+pressure"+ccn)

*We added the sentence "All in-situ measurements are converted to under ambient temperature and pressure." In line 162-163 of the revised manuscript.*

\* Please refrain from using subjective statements regarding easiness of further code modifications (unless depicting such developments with examples):
- remove "easily" in the abstract (p1/l27, also suggest removing last sentence of the abstract)

*Revised as suggested*
- remove "Only minimal modifications to the diagnostics package are needed ..." (p4/l102-102)

*Removed as suggested*
- remove "It is relatively straightforward to add ..." (p4/l114-115)

*Removed as suggested*
- rephrase "can be easily implemented" (p30/l525)

*This sentence is revised as "The code structure is designed to be flexible and modular for future extension to other field campaigns or additional datasets."*

- remove "It can also be applied to evaluate other ESMs with small modifications" (p30/l540)

*This sentence is revised as "It can also be applied to evaluate other ESMs with necessary modifications to fit different model output formats."*

* Whenever referring to 25%-75% percentiles, perhaps worth to use the notion of "inter-quartile range"

*We replace some of the "25%-75% percentiles" with "inter-quartile range" (lines 255, 270, 277, 358 in the revised manuscript), but keep those when they appear with 5% and 95% percentiles for consistency.*

* unclear statement: "aerosol properties decrease with height" (p17/l325-326)

*This sentence is revised as "The observed aerosol concentrations of number and chemical composition decrease with height"*

Hope that helps,
Best regards,
Sylwester Arabas